# Global morphogenetic flow is accurately predicted by the spatial distribution of myosin motors

Sebastian J Streichan[1,2]*, Matthew F Lefebvre[3], Nicholas Noll[2], Eric F Wieschaus[3,4], Boris I Shraiman[1,2]*

[1]Kavli Institute of Theoretical Physics, University of California, Santa Barbara, United States; [2]Department of Physics, University of California, Santa Barbara, United States; [3]Department of Molecular Biology, Princeton University, Princeton, United States; [4]Howard Hughes Medical Institute, Princeton University, Princeton, United States

**Abstract** During embryogenesis tissue layers undergo morphogenetic flow rearranging and folding into specific shapes. While developmental biology has identified key genes and local cellular processes, global coordination of tissue remodeling at the organ scale remains unclear. Here, we combine *in toto* light-sheet microscopy of the *Drosophila* embryo with quantitative analysis and physical modeling to relate cellular flow with the patterns of force generation during the gastrulation process. We find that the complex spatio-temporal flow pattern can be predicted from the measured meso-scale myosin density and anisotropy using a simple, effective viscous model of the tissue, achieving close to 90% accuracy with one time dependent and two constant parameters. Our analysis uncovers the importance of a) spatial modulation of myosin distribution on the scale of the embryo and b) the non-locality of its effect due to mechanical interaction of cells, demonstrating the need for the global perspective in the study of morphogenetic flow.
DOI: https://doi.org/10.7554/eLife.27454.001

*For correspondence:
streicha@kitp.ucsb.edu (SJS);
shraiman@kitp.ucsb.edu (BIS)

Competing interests: The authors declare that no competing interests exist.

## Introduction

Animal development is characterized by highly dynamic rearrangements of mechanically coupled cells. Such rearrangements must be tightly coordinated across the embryo to achieve normal morphology and organogenesis. During gastrulation of *Drosophila*, for example, the embryonic blastoderm – an epithelial monolayer of about 6000 cells on the surface of the embryo – undergoes a dramatic deformation that changes tissue topology and gives rise to the three germ layers. These processes involve a coherent flow of cells along the surface of the epithelial monolayer, which in turn drives folding and defines future shape of the embryo. The most prominent aspects of gastrulation are the formation of the ventral furrow which initiates the invagination and internalization of the mesoderm (*Martin et al., 2009*), and germ-band extension which involves convergent extension of the lateral ectoderm and the flow of the ventral germ-band onto the dorsal side of the embryo (*Leptin, 1995*). Both of these processes have been extensively studied, leading to the identification of developmental patterning genes specifically required for each process (*Irvine and Wieschaus, 1994*). Live imaging has also uncovered process-specific cell behaviors such as apical constriction of presumptive mesoderm cells during ventral furrow formation (*Martin et al., 2009*) and intercalation of neighboring cells in the lateral ectoderm during convergent extension (*Zallen and Wieschaus, 2004*; *Bertet et al., 2004*). These behaviors are associated with localized activity of the force generating non-muscle myosin II (*Martin et al., 2009*; *Irvine and Wieschaus, 1994*; *Zallen and Wieschaus, 2004*; *Bertet et al., 2004*). However, despite considerable understanding of the local

processes involved in such cellular rearrangements, a coherent picture of global morphogenetic flows has remained elusive (*Butler et al., 2009*; *Lye et al., 2015*; *Blanchard et al., 2009*).

Understanding how cell flows are coordinated across different cell populations requires distinguishing the roles of local cell behavior and long-range intercellular interactions. To what extent is the transformation of tissue driven locally by the processes associated with cells at that position? How important is the long-range interaction between different regions? In the context of the fly embryo, VF formation seems well explained locally by the apical area contraction of ventral mesoderm cells (*Martin et al., 2009*). On the other hand, the non-local interactions between the VF constriction (or posterior midgut invagination [*Collinet et al., 2015*]) and the convergent extension of lateral ectoderm remain a subject of active study (*Collinet et al., 2015*; *Rauzi et al., 2015*; *Rickoll and Counce, 1980*) which requires quantitative multi-scale analysis.

There are two complementary approaches towards quantitative analysis of tissue flow. One approach focuses on cell-scale behavior aiming to decompose tissue flow into specific cellular processes such as cell-shape change and intercalation (*Etournay et al., 2015*; *Bosveld et al., 2012*). Alternatively one can 'zoom out', taking a continuum mechanics approach that aims to describe tissue flow on the whole organ scale (*Landau et al., 2012*; *Prost et al., 2015*; *Marchetti et al., 2013*). This coarse-grained mesoscopic perspective captures correlations in cell behavior which stem from intercellular interactions and the supra-cellular organization (*Martin et al., 2009*; *Blankenship et al., 2006*) of the cytoskeleton in epithelial tissues. In biophysics, the continuum mechanics approach has been developed to understand the behavior of active gels (*Prost et al., 2015*; *Marchetti et al., 2013*) during myosin driven viscoelastic flow (*Mayer et al., 2010*; *Behrndt et al., 2012*) and has been successfully used to model cortical flows in *C. elegans* zygotes at the first-cleavage state (*Naganathan et al., 2014*). Recent theoretical work (*Noll et al., 2017*) provides a bridge between cell-based and meso-scale continuum descriptions, focusing on the non-trivial consequences of stress-dependent active cytoskeletal processes. Here, we shall use continuum mechanics approach to set up a framework for predicting global tissue flow at the whole organ level.

The main advantage of the continuum mechanics approach is its ability to capture key aspects of force balance associated with local deformation and flow. It allows to describe quantitatively, with only a few parameters, how the effect of local forcing spreads across a tissue. The tendency of cells to stick together and resist deformation results in a non-local relation between the myosin activity that drives the flow and actual flow velocities. The continuum mechanics approach therefore enables one to test different hypotheses helping to identify key contributing processes. For example, the question of local versus non-local response in the continuum mechanics approach translates into specific hypotheses regarded force balance: are the myosin-generated forces balanced locally by traction relative to a substrate or do they propagate within the epithelium layer through cell deformation and viscous coupling? Quantitative analysis can then be used to build, starting from the simplest model, a sequence of approximations that capture biological reality in increasing detail.

We shall describe below a novel image analysis-based approach that will use continuum mechanics to quantitatively relate different observables and will show that myosin distribution and anisotropy on mesoscopic scale is a fully adequate proxy of physical stress, thereby enabling a surprisingly predictive description of global flow. Specifically, we shall show that embryo-scale tissue transformations during *Drosophila* gastrulation are represented by a temporal sequence of three topologically distinct flow field configurations. Each phase is accompanied by a characteristic spatial distribution of myosin molecular motors both on the basal as well as apical cell surface, which we quantify in terms of a coarse-grained 'myosin tensor' that captures both myosin concentration and anisotropy. To relate the observed global flow fields to myosin apical and basal distributions we assume that tissue flow is driven by stress proportional to the myosin tensor, and is effectively viscous with two parameters: effective shear and bulk viscosities, the latter controlling the compressible component of the flow. With a total of just three global parameters (only one of them time dependent), this simple model achieves remarkable agreement between predicted and measured spatio-temporal pattern of the flow. The analysis uncovers the importance of a) spatial modulation of myosin distribution and b) the long-range spreading of its effect due to mechanical interaction of cells. In particular, we find that transition to the germband extension phase of the flow is associated with the onset of effective areal incompressibility of the epithelium, which makes the relation of the flow and myosin forcing strongly non-local. Our quantitative analysis also reveals a new function for basal myosin in generating a dorsally directed flow and, combined with mutant analysis, points to an unconventional

control mechanism of this function through *twist* -dependent reduction of basal myosin levels on the ventral side. Finally, we shall argue that the ability to quantitatively describe the relation between the flow and its myosin-generated forcing provides a new approach to the study of the processes that control morphogenetic transformations, which can be used to disentangle novel control mechanisms such as mechanical feedback from the effects of gene expression patterning.

## Results

To enable our study, we generated a pipeline that combines *in toto* light sheet microscopy (*Krzic et al., 2012*; *Tomer et al., 2012*) (*Figure 1a*), tissue cartography (*Heemskerk and Streichan, 2015*) (*Figure 1b*), and segmentation-free anisotropy detection to quantify global tissue flows, and myosin activation patterns (*Figure 1—figure supplements 1–4*). Using optical flow velocimetry applied to cylinder projections of the Surface of Interest (SOI) passing through cells below the apical cell surface (see SI for details *Figure 1—figure supplement 5c*), we find that tissue remodeling during *Drosophila* gastrulation is characterized by three simple flow field configurations (*Figure 1c–e*). The earliest flows start well before the ventral furrow (VF) forms, and are characterized by a dorsal sink and ventral source (*Figure 1c*). In contrast to the VF, no cells are internalized during this flow, but rather cells reduce cross section on the dorsal side (*Figure 1—figure supplement 1c*). As the VF forms, source and sink swap sides and a large group of cells internalize on the ventral side, as mesoderm precursors leave the surface of the blastoderm (*Figure 1d*). During germband extension (GBE), the flow pattern exhibits two saddles arranged on the dorsal and ventral sides as well as four vortices, two in the posterior and two in the anterior end (*Figure 1e*). Each of the three flow fields is accompanied by a typical spatial myosin configuration. The pre-VF flow associates with basal myosin that exhibits a pronounced Dorso Ventral (DV) asymmetry (*Warn et al., 1980*; *Sokac and Wieschaus, 2008*; *Polyakov et al., 2014*), with high levels of myosin on the dorsal and low levels on the ventral side (*Figure 1f*), while the apical pool appears uniform across the surface (*Figure 1i*, *Figure 1—figure supplement 5d–g*). The basal pool remains asymmetric during VF flow (*Figure 1g*), but the apical pool now also develops DV asymmetry in reversed orientation (*Figure 1j*). The asymmetry on the apical surface becomes further pronounced in the GBE-phase (*Figure 1h,k*, *Figure 1—figure supplements 6* and *7*).

Global changes in myosin pools are a hallmark for transitions in flow field configuration (*Figure 2a*, *Figure 1—figure supplements 1*, *2*, *3* and *4*). Myosin is initially enriched in the basal pool, and as sink and source swap position, it begins to accumulate on the apical side. While the basal pool is isotropic (*Figure 2—figure supplement 4a*), cortical myosin on the apical cell surface is known to polarize during convergent extension (*Zallen and Wieschaus, 2004*; *Bertet et al., 2004*). To quantify this effect at the tissue level, we developed an automated segmentation-free anisotropy detection algorithm (*Figure 2b*, *Figure 2—figure supplement 3a,b*). Available methods for anisotropy detection mostly operate at the single cell level and construct a nematic tensor by integrating signal intensities along cell outlines (*Aigouy et al., 2010*). At the organismal scale membrane segmentation is costly, and often fails to define closed outlines of cells using only a polarized membrane marker. We overcome the need for fiduciary markers that increase experimental complexity by shifting the perspective to cell edges and using the Radon transform to implement a robust and rapid segmentation-free algorithm for detecting course-grained anisotropy (*Figure 2b*) (*Radon, 1917*). Radon transforms integrate signal along lines of given orientation and normal distance from the origin. In this way, edges are mapped to peaks that reflect the total intensity along the length of an edge (*Figure 2b*) (see Appendix 1 for detail). Edge orientation and average myosin intensity are described by a $2 \times 2$ symmetric matrix (of rank one) defining the local 'myosin tensor' (*Figure 2b*). By averaging the resulting tensors in a given region, we obtain a quantitative description of local tissue anisotropy and overall levels that reflects the intensity-weighted average of cell edges. The resulting course-grained tensor has a non-zero trace, and thus can be separated into an isotropic and a traceless anisotropic part (*Figure 3a*, *Figure 2—figure supplement 3b*), from which we construct a measure for anisotropy (which is low in the basal pool *Figure 2—figure supplement 4a*). The anisotropic signal in the apical pool starts out low, but increases from about 8 min corresponding to late stage 6 (*Figure 2a*, *Figure 2—figure supplement 4b,c*). The anisotropy axis, readily computed by the eigenvectors of the myosin tensor, aligns well with local tangent to pair rule gene expression boundaries (*Figure 2a,d*, *Figure 2—figure supplement 4d*). This is the expected result

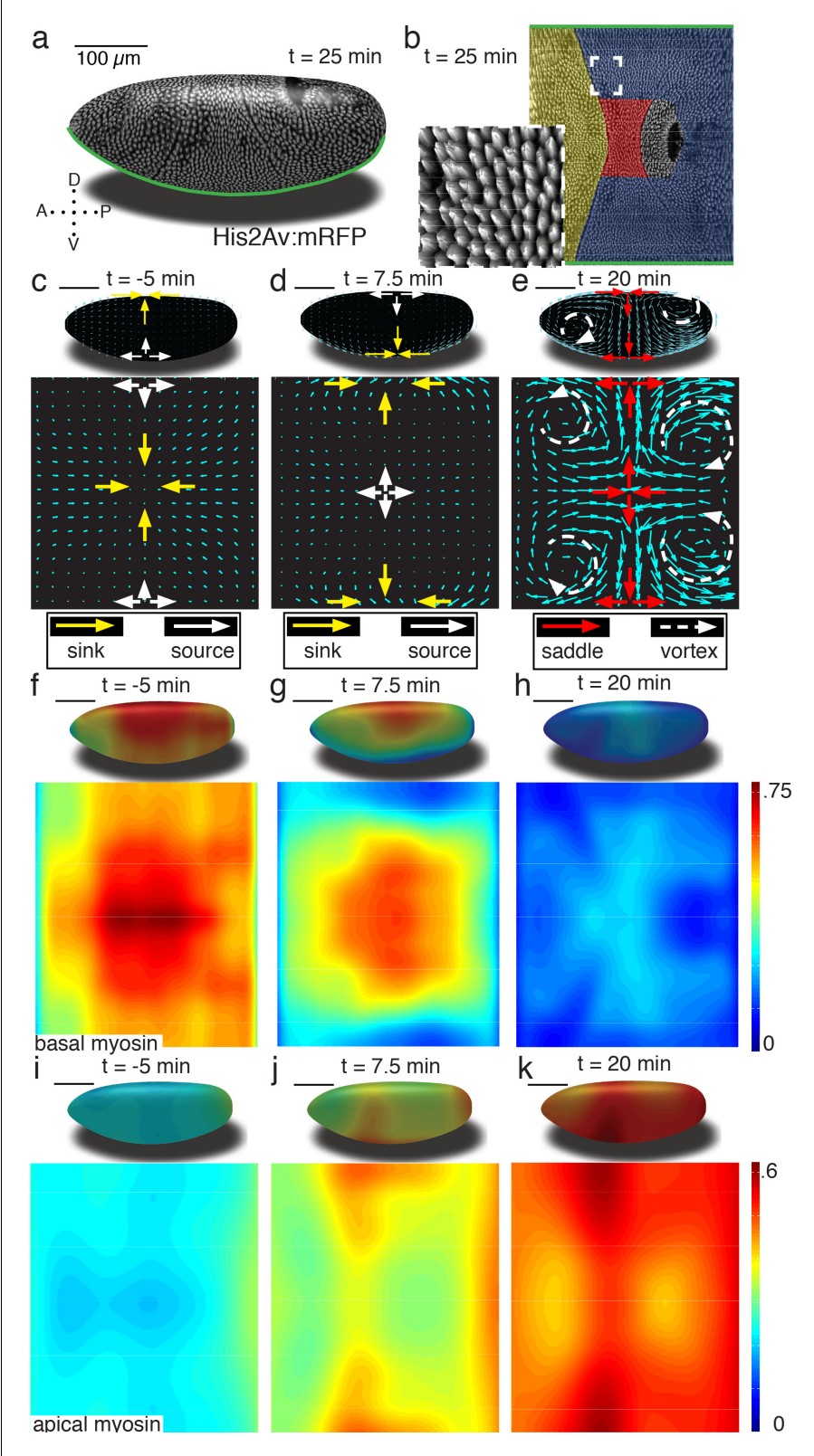

**Figure 1.** Tissue deformations of *Drosophila melanogaster* embryos during gastrulation, captured by three simple flow fields. (a) Stage 7 embryo labeled with His2Av:mRFP. Anterior is to the left, dorsal up. Time is chosen such that 0 min coincides with the first occurrence of the cephalic furrow (CF). All scale bars indicate 100 μm. (b) Thin (midplane, *Figure 1—figure supplement 1*) layer through embryo shown in (a), with prospective head, germband and amnioserosa color-coded. Anterior is to the left, posterior to the right, dorsal is in the center and ventral is on top and bottom. Inset
*Figure 1 continued on next page*

*Figure 1 continued*

shows zoom into anterior germband region. (**c–e**) Flow field on 2D projections for representative time points of the pre-Ventral Furrow (pre-VF) phase (**c**), Ventral Furrow (VF) phase (**d**), and germband phase (GBE) (**e**). Cyan arrows indicate tissue flow field. Bold arrows indicate flow field topology: sinks (yellow), sources (white), saddles (red) and vortices (dashed white). Insets show flow field on corresponding 3D surface. (**f–h**) Normalized myosin distribution on basal cell surface corresponding to times shown in (**c–e**). Color code from lowest 0 to highest 1. (**i–k**) As (**f–h**) except for isotropic pool on apical cell surface.

DOI: https://doi.org/10.7554/eLife.27454.002

The following figure supplements are available for figure 1:

**Figure supplement 1.** Definition of embryo shape and relevant surfaces of interest.

DOI: https://doi.org/10.7554/eLife.27454.003

**Figure supplement 2.** Myosin timecourse on the apical surface.

DOI: https://doi.org/10.7554/eLife.27454.004

**Figure supplement 3.** Time course of Myosin on basal surface, as described in *Figure 1—figure supplement 1g*.

DOI: https://doi.org/10.7554/eLife.27454.005

**Figure supplement 4.** Magnified view of time course of Myosin on basal surface, as described in *Figure 1—figure supplement 1e,g*.

DOI: https://doi.org/10.7554/eLife.27454.006

**Figure supplement 5.** Quantitative analysis of ensemble flow field.

DOI: https://doi.org/10.7554/eLife.27454.007

**Figure supplement 6.** Isotropic basal myosin, (as main text *Figure 1g*).

DOI: https://doi.org/10.7554/eLife.27454.008

**Figure supplement 7.** Basal myosin quantification light sheet versus confocal.

DOI: https://doi.org/10.7554/eLife.27454.009

given that anisotropies are thought to be driven by the patterned juxtaposition of pair-rule gene expression (*Zallen and Wieschaus, 2004*).

We have examined and quantified tissue flow and myosin distribution in multiple (N = 22) wild-type embryos and found it highly reproducible (*Figure 1—figure supplement 5*). For the purpose of quantitative analysis presented below, we shall use the (suitably aligned) 'ensemble'-averaged flow and myosin distribution (see SI).

To relate myosin to stress, we assume signal intensity is proportional to myosin motor concentration and its local activity. The latter – pulling on cytoskeletal actin filaments – generates local force dipoles, which can be explicitly described in terms of local stress tensor (see Appendix for details) (*Prost et al., 2015*; *Marchetti et al., 2013*). On the coarse grained level, resulting stress would be defined by the activity weighted average over filament orientations and hence proportional to the myosin tensor as we define it. The resulting force per unit area of the epithelial layer is then proportional to the divergence of the myosin tensor (*Landau et al., 2012*). Note that the isotropic component of the myosin distribution (observed both in the apical and the basal pool) also generates a force that is proportional to the gradient of the measured concentration intensity profile (*Figure 1f–k*).

To relate myosin generated stress to morphogenetic flow, we assume that on the mesoscopic scale tissue flow is governed by effective viscoelasticity which arises from the mechanical properties of the underlying cytoskeletal network within the two dimensional epithelial layer of cells. This model assumes that on short time scales tissues respond elastically to mechanical perturbations (*Bambardekar et al., 2015*), yet on longer time scales elastic stress is relaxed through active rearrangement of the cytoskeleton as cells adapt to the imposed deformation. On the longer time scale tissue dynamics can be described by a two-dimensional effective viscous flow equation with two *effective viscosity* parameters that (see Appendix 2) are directly related to the two elastic constants: shear modulus (controlling 'sliding' of cells relative to each other) and the planar bulk modulus (controlling areal compression or dilation) (*Prost et al., 2015*; *Marchetti et al., 2013*; *Martin et al., 1972*) (*Figure 3a*, *Figure 3—figure supplement 1b*). We note that effective viscosity spreads the impact of local forcing, generating a non-local response so the flow at any given point integrates the influence of forces acting all over the embryo. Inverting the equation using the finite element method, we obtain a quantitative prediction for the flow field generated by measured myosin localization patterns (see SI for details). Our model has only three global parameters: the ratio of effective viscosities, and the conversion factors relating normalized apical and basal myosin intensity to stress

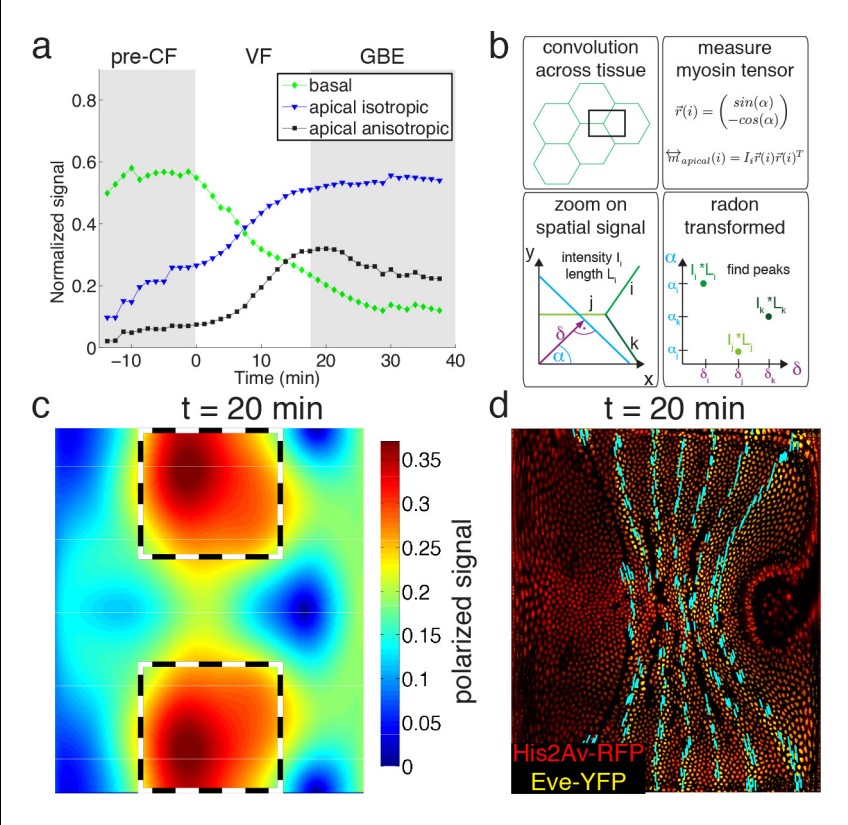

**Figure 2.** Quantitative analysis of myosin distribution and anisotropy reveals transition across pools. (**a**) Normalized signal strength of basal, apical, and polarized pools over time in the lateral ectoderm (outlined as dashed box in c). First gray shaded box at t < 0 min indicates times before CF formation (pre-CF), second shaded indicates GBE. (**b**) Automated extraction of polarization based on images, and quantitative summary as nematic tensor. Top left box shows cell outlines in part of a tissue, and a region of interest (ROI), that moves across the tissue. Bottom left box shows zoom on spatial signal in ROI. Colors indicate potentially different intensities of lines labeled i,j,k. Average intensity and length of lines in images are denoted I and L respectively. Radon transforms integrate signal along lines (cyan) of orientation α at normal-distance δ from the origin (purple). Bottom right inset shows sketch of resulting Radon-transformed signal. Note that lines are peaks at angle α, and distance δ, of height L*I after transformation. Top right inset shows definition of unit vector with orientation of edge i. Definition of local myosin tensor (only computed on apical surface, see *Figure 2—figure supplement 4*) for edge *i* is obtained by contracting unit edge vector with itself and weighted by line average intensity. (**c**) Magnitude of myosin anisotropy on pullback (see SI for definition). Dashed box indicates region of interest used to compute time traces in a. (**d**) Axis of myosin anisotropy (in cyan) overlaid on embryo labeled with his2Av-RFP in red, and eve-YFP in yellow. For simplicity of comparison, the field is only shown along even skipped stripes. For more detailed analysis see *Figure 2—figure supplement 3f*.

DOI: https://doi.org/10.7554/eLife.27454.010

The following figure supplements are available for figure 2:

**Figure supplement 1.** Illustration of how to construct a Radon transform for an image with constant background $I = 0$, shown in black and foreground $I = 1$, shown in white (top), and resulting radon transform (bottom).

DOI: https://doi.org/10.7554/eLife.27454.011

**Figure supplement 2.** Example of edges identified with our anisotropy detection algorithm, and a magnification in a region of interest showing result in comparison with underlying raw data (left).

DOI: https://doi.org/10.7554/eLife.27454.012

**Figure supplement 3.** Continuous representation of myosin tensor on the mesoscale.

DOI: https://doi.org/10.7554/eLife.27454.013

**Figure supplement 4.** Myosin tensor on the mesoscale.

DOI: https://doi.org/10.7554/eLife.27454.014

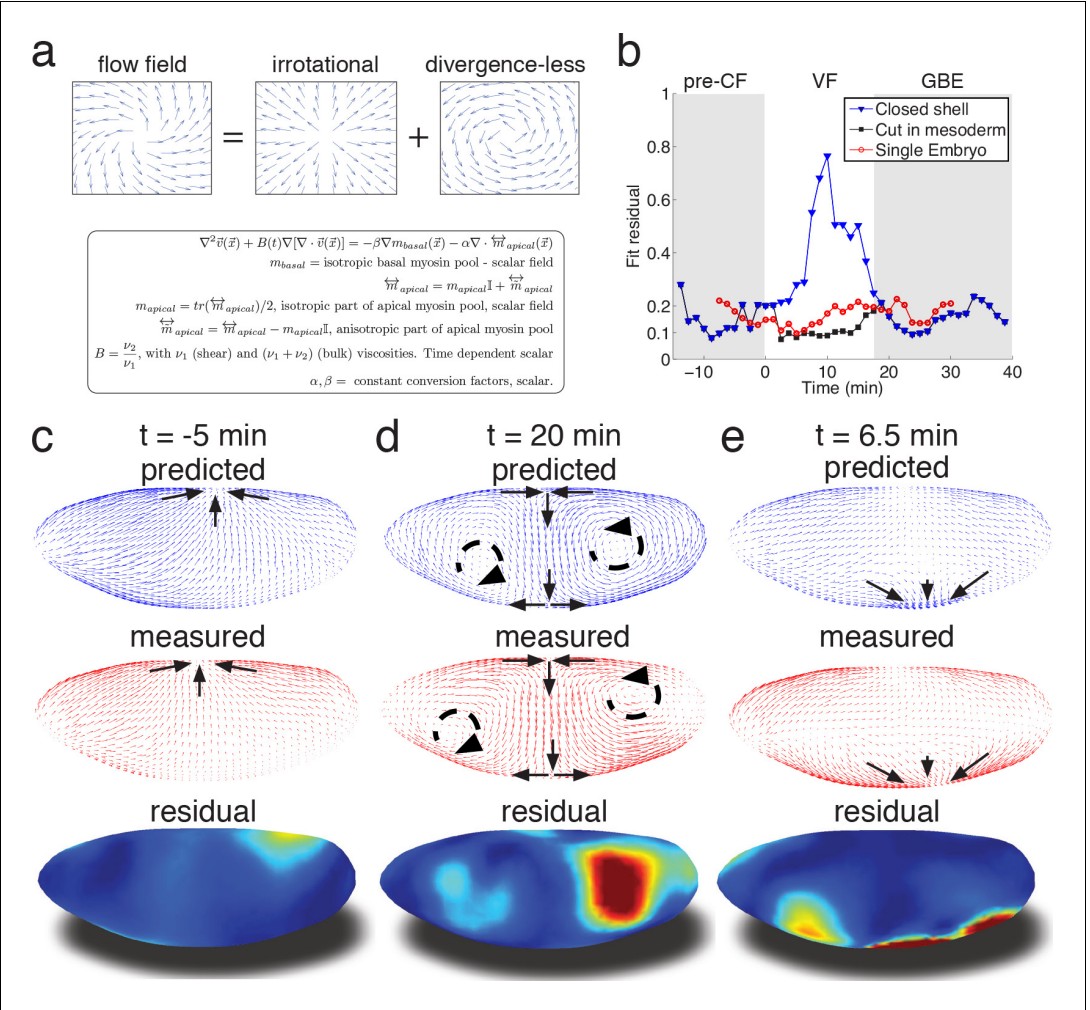

**Figure 3.** Biophysical model quantitatively predicts tissue flow based on quantitative measurements of myosin distribution. (a) Proposed mathematical description of the flow parameterizes complex mechanics of cytoskeleton in terms of the shear $\nu_1$ and $\nu_2$ bulk effective viscosities. The flow is driven by the force proportional to the divergence of the myosin tensor (see SI) on the right-hand-side of Equation 3a. Because effective viscosity tends to suppress velocity differences of neighboring cells, the response to local forcing is felt globally, e.g. effect of a local myosin perturbation results in local as well as non-local changes of the flow field. (b) Fit residual, comparing predicted flow field with measured flow field (see SI Finite Element implementation for a detailed definition of the residual) as a function of time. Both fields are normalized for average magnitude. The average magnitude of predicted velocity field defines one of our fitting parameters. Images of the single embryo are shown in *Figure 1—figure supplements 2–3* (c–e) Representative time points of morphogenetic flow: pre-CF (c), GBE (d) and VF (e). From top to bottom: spatial distribution of predicted (blue), measured (red) flow field, and residual (blue best agreement, red worst, on a scale from 0 to 1). For the case of VF flow, predictive model is modified to allow for a 'cut' in ventral region (see SI text, and *Figure 3—figure supplement 1* for detail).
DOI: https://doi.org/10.7554/eLife.27454.015

The following figure supplement is available for figure 3:

**Figure supplement 1.** Finite element realization of model and parameter values.
DOI: https://doi.org/10.7554/eLife.27454.016

(*Figure 3—figure supplement 1c*). To keep the model as simple as possible, we do not allow spatial dependence of these parameters and keep conversion factors constant, leaving the ratio of viscosities as the only time-dependent global fitting parameter (*Figure 3—figure supplement 1b*).

Even without spatial modulation of the parameters, the model achieves about 90% accurate description of the flow pattern before and after VF invagination (see *Figure 3b*). The main

discrepancy of model predictions for pre-VF flow (see *Figure 3c*) is a displacement of sink and source positions along the AP axis by less than 30 μm. Prediction of GBE flow essentially agrees with measurements across the entire embryo, with the exception of a domain close to the vortices on the posterior end, due to a mismatch of fixed-point location (*Figure 3d*). Remarkably, our model is even able to correctly predict subtle differences between anterior and posterior fixed points along the DV axis (*Figure 3d*). Measured flow is first dominated by sources and sinks that disappear later during GBE, suggesting that before and during VF invagination cells are less resistant to surface area compression than during GBE. Indeed, quantitative comparison with an independently measured flow field (*Figure 1c–e*) shows that the $v_{20}/v_{27}$ ratio increases dramatically at the start of GBE phase (corresponding to the relative increase of the underlying 2D bulk modulus, see Appendix 2, *Figure 3—figure supplement 1b*) resulting in effective incompressibility of apical surface of cells. The temporal coincidence between completion of cellularization and increase of the bulk modulus provides an intriguing possible explanation of how the continuous transition in our time dependent variable might be realized. Poor agreement during VF invagination is due to a significant fraction of cells internalizing and thus leaving the surface. To account for this effect, we extend the model to allow a 'cut' in the lattice along ventral midline with an imposed in-plane boundary force (perpendicular to the cut) representing the pulling effect of the VF (see SI for detail, *Figure 3—figure supplement 1a*). This relatively simple extension allows to recover ~90% accuracy (*Figure 3be*), illustrating how regional inhomogeneity associated with particular morphogenetic events could be quantitatively captured by suitable generalizations.

To evaluate the fit obtained in wild type embryos, we examined flows in mutant embryos in which the distribution of myosin is altered. Analysis based on tissue tectonics (*Blanchard et al., 2009*) has shown that strain rates in *twist (twi)* embryos, which lack the VF, exhibit slower kinetics compared to WT (*Butler et al., 2009*), however, the cause of this remains a subject of debate (*Butler et al., 2009*; *Lye et al., 2015*; *Collinet et al., 2015*). We have quantified the flow field and myosin activity patterns in *twi* mutants (*Figure 4—figure supplement 1*), and find that our model is able to accurately predict the flow profiles (*Figure 4a*). During early flow phases – corresponding to times of pre-VF flow in WT – DV asymmetry of the basal myosin pool is strongly reduced in comparison to WT, as is tissue movement towards the dorsal pole (*Figure 4b*, *Figure 1—figure supplement 1*, *Figure 4—figure supplement 1a,d*). Moreover, anisotropy of the apical myosin pool increases at a slower rate as compared to WT. As previously reported for strain rates (*Butler et al., 2009*), this is most pronounced for the first 20 min (*Figure 4c*, compare *Figure 4* – fig. supplement = 0 with *Figure 2—figure supplement 3e,f*). In *bcd nos tsl (bnt)* embryos lacking all AP patterning, the early basal DV asymmetry is similar to WT, with only slightly reduced myosin asymmetries and dorsal movement (*Figure 4b*, *Figure 2—figure supplement 1*). At later stages, however, anisotropy of the apical myosin pool remains low and comparable to pre-VF WT levels. This result is expected given the uniform expression of pair-rule genes in a *bnt* genetic background (*Blankenship et al., 2006*) (*Figure 4c*). Consistent with these myosin distributions, we see the early dorsal flow associated with basal myosin asymmetry but a failure to produce the more complex later flow patterns with their characteristic saddles and vortices. On a quantitative level our model's predictive power for AP patterning deficient *bnt* mutant embryos is comparable to WT and *twi* mutants (*Figure 4a*).

## Discussion

In summary, we have presented a simple biophysical model of morphogenetic flow that quantitatively describes complex tissue motion in terms of a hydrodynamic equation parameterized by two effective viscosities. The flow is driven by the stress defined by a linear superposition of two myosin tensors describing the apical and basal myosin pools. We propose that the basal myosin pool forms an isotropic and contiguous network (*He et al., 2016*), contracting in a similar fashion as purified actomyosin gels in vitro (*Bendix et al., 2008*; *Alvarado et al., 2013*). Imbalance within this network, caused by the *twi* dependent depletion of myosin on the ventral side, drives global dorsal-ward flow in the pre-VF phase, which continues to contribute until early GBE (*Figure 4d*). Interestingly, in silico perturbations indicate the local depletion (on the ventral side) has a global effect, most evidently manifested by a 'sink' on the dorsal side, which is lost in a simulation using the same parameters as WT, but no DV modulation of the basal myosin pool (*Figure 4d*, left panels). The apical pool decomposes into isotropic and anisotropic components. In addition to previously described accumulation

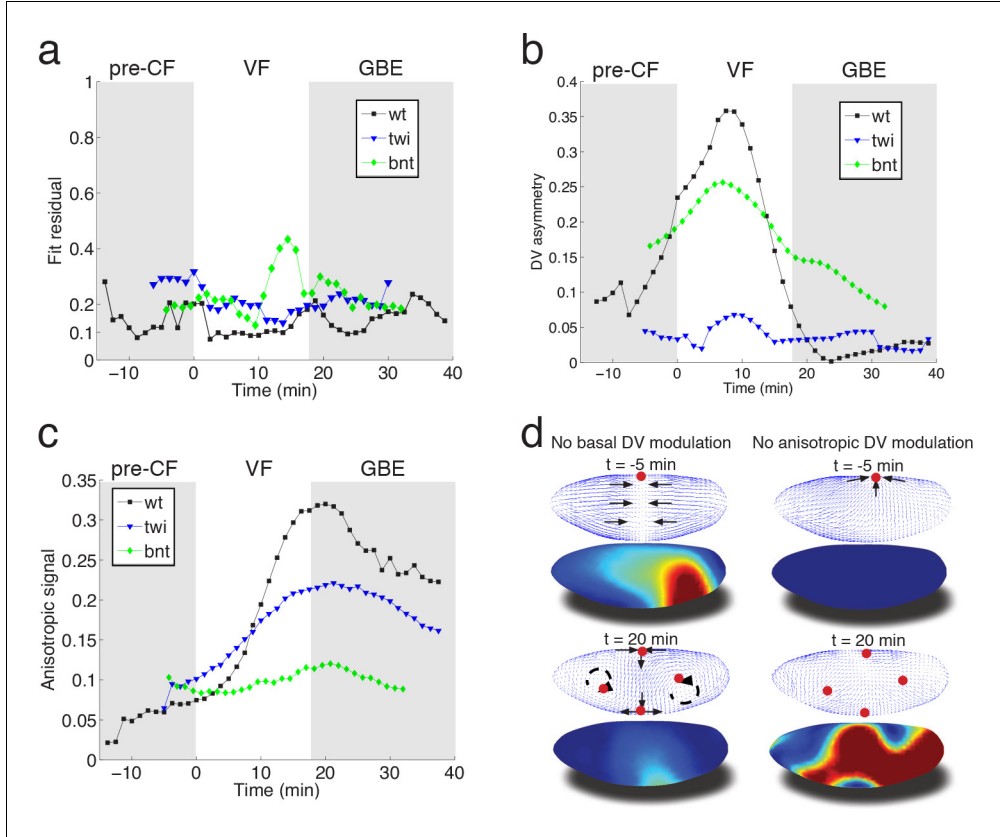

**Figure 4.** Mutant analysis reveals global modifications of myosin dynamics. (**a**) Fit residual as in *Figure 3b*, for *twi*, and *bcd nos tsl* mutants (7, and 7 embryos in ensemble). WT is shown as reference. (**b**) Amplitude of basal myosin pool along DV axis for WT and mutants in (**a**). (**c**) Polarized apical myosin in mutants shown in (**a**) as function of time. (**d**) Theoretical comparison of DV constant basal pool (i.e. no gradient in DV direction) (left column), or DV constant anisotropic apical pool (i.e. no gradient in DV direction) (right column) with predicted flow based on full myosin tensor (compare to *Figure 3c,d* respectively). Black arrows indicate flow field topology, and red dots the fixed point from prediction based off of full myosin tensor. Model parameters are the same as previously determined for the WT (*Figure 3—figure supplement 1*).

DOI: https://doi.org/10.7554/eLife.27454.017

The following figure supplement is available for figure 4:

**Figure supplement 1.** Twist mutant flow field and myosin analysis.
DOI: https://doi.org/10.7554/eLife.27454.018

of isotropic myosin in ventral regions (*Martin et al., 2009*), we observe a striking gradient of anisotropic apical myosin along the DV axis, reaching highest levels in lateral ectoderm and lowest levels in amnioserosa tissue at the dorsal pole. Because the force driving the flow arises from the non-uniformity of the stress, this modulation of myosin distribution is critical for the dynamics. While local rate of cell intercalation is often interpreted in terms of local myosin distribution on cellular and sub-cellular scales, our model shows that the local rate of strain is a result of the tissue-wide distribution of forces generated by the spatial non-uniformity of myosin (mathematically described by the divergence of the myosin tensor) (*Figure 4d*, right panels). The importance of spatial modulation (*Figure 4d*) suggests a novel role of the *dorsal* signaling pathway in generation of GBE flow. Surprisingly, in *twi* mutants both the rate of increase as well as the peak myosin anisotropy are significantly reduced in the first 20 min of GBE flow (*Figure 4c*, *Figure 4—figure supplement 1*). The reduced intercalation and strain rates observed in these mutants has been previously reported (*Butler et al., 2009*), and interpreted in terms of possible generation of AP forces by the internalized VF (absent in *twi* mutants). Our model accounts for the reduced rate of strain in terms of the changes in spatial distribution and the reduced level of myosin anisotropy. This however brings up the question of how

elimination of *twi* expression in the ventral mesoderm affects myosin anisotropy in the lateral ectoderm. We suggest that this effect may be due to mechanical feedback on myosin recruitment, which relates the later to local rate of strain. Through this 'dynamic recruitment' effect (*Noll et al., 2017*; *Fernandez-Gonzalez et al., 2009*), changes in the ventral region that modify global flow patterns can affect myosin distribution and anisotropy in the lateral region. In this way, local modification of the myosin pattern can produce not only a non-local perturbation of the flow, but also a non-local perturbation of myosin distribution. The global nature of the flow is reinforced by the observed transition towards areal incompressibility at the onset of GBE-flow, which together with reduced polarization kinetics and reduced strain rates observed in *twi* mutants, indicates that non-local consequences of stress generated largely in lateral ectoderm can account for the dorsal movement of the posterior midgut.

Taken together, our observations show that morphogenetic flow is a global response to local forcing which arises from the spatial modulation of myosin density and anisotropy. The latter is derived from the spatial patterns of developmental transcription factors, but we suggest may also involve mechanical feedback affecting recruitment of myosin. Our quantitative approach provides a framework for integrating the effect of local factors in the description of the global flow.

## Materials and methods

### Fly lines used

His2Av-mCherry (*Krzic et al., 2012*), bcd$^{e1}$nos$^{bn}$tsl$^4$/TM3, halo twi$^{ID96}$/Cyo (twi$^{ID96}$ is also known as twi [*Martin et al., 2009*]), sqh-GFP klar (*Martin et al., 2009*), OregonR. Embryos where dechorionated following standard procedures, and mounted in agarose gels as previously described (*Krzic et al., 2012*).

### Light sheet microscopy

Fluorescence-based live imaging was carried out on a MuVI SPIM (*Krzic et al., 2012*). Briefly, the optics consisted of two detection and illumination arms. Each detection arm forms a water-dipping epifluorescence microscope, consisting of an objective (Apo LWD 25x, NA 1.1, Nikon Instruments Inc.), a filter wheel (HS-1032, Finger Lakes Instrumentation LLC), with emission filters (BLP01-488R-25, BLP02-561R-25, Semrock Inc.), tube lens (200 mm, Nikon Instruments Inc.), and an sCMOS camera (Zyla 4.2, Andor Technology plc.), with an effective pixel size of 0.26 µm. Each illumination arm consisted of a water-dipping objective (CFI Plan Fluor 10x, NA 0.3), a tube lens (200 mm, both Nikon Instruments Inc.), a scan lens (S4LFT0061/065, Sill optics GmbH and Co. KG), and a galvanometric scanner (6215 hr, Cambridge Technology Inc.), fed by lasers (06-MLD 488 nm, Cobolt AB, and 561LS OBIS 561 nm, Coherent Inc.). Optical sectioning is achieved by translating the sample using a linear piezo stage (P-629.1cd with E-753 controller) sample rotation is performed with a rotational piezo stage (U-628.03 with C-867 controller) and a linear actuator (M-231.17 with C-863 controller, all Physik Instrumente GmbH and Co. KG).

### Experiment control and data fusion

Stages and cameras are controlled using Micro Manager (*Edelstein et al., 2014*), to coordinate time-lapse experiments, running on a Super Micro 7047GR-TF Server, with 12 Core Intel Xeon 2.5 GHz, 64 GB PC3 RAM, and hardware Raid 0 with 7 2.0 TB SATA hard drives. Samples were recorded from two, by 90$^0$ rotated views, at a typical optical sectioning of 1 µm, and temporal resolution of 75 s. As previously described (*Krzic et al., 2012*), MuVI SPIM optical stability allows a fusion strategy based on a diagnostic specimen. Recorded once per experiment, the diagnostic specimen is used to determine an initial guess for an affine transformation, which we feed into a rigid image registration algorithm (*Klein et al., 2010*), to fuse individual views, resulting in an isotropic resolution of. 26 µm in the registered image.

### Surface of Interest extraction

We used tissue cartography to extract surfaces of interest (SOI) from embryos (*Heemskerk and Streichan, 2015*). Briefly, we identify the outline of the sample using the Ilastik detector to determine a point cloud for SOI construction. In a fitting step (implemented in the sphere-like fitter), we

create a smooth description of the SOI in terms of cylinder coordinates defined by AP axis and azimuth (*Heemskerk and Streichan, 2015*). Image intensity data are then projected onto a nested group of 5 layers (two normally evolved layers above and below the SOI), each three pixels apart, defining a 3–4 µm thick 'curved image stack'. For analysis the maximum intensity projection of nested layers was used. Although the shape of embryos of the same genotype is highly reproducible, small differences in the underlying point cloud can result in small differences of the SOI passing through the apical cell surface. To simplify comparison between embryos, we create a standard projection on a cylinder grid of fixed size, with the embryo surface oriented such that apical is left, posterior right, dorsal in the center, and ventral on top and bottom (*Figure 1*). Systematic distortions of measurements due to projecting the curved embryo surface to the plane are corrected using the metric tensor (*Heemskerk and Streichan, 2015*).

The apical surface is static, while the dynamic basal cell surface moves with the cellularization front. Projections of the latter could be created by reading signal on a surface obtained by evolving the apical SOI along its normal basal wards. However, small differences in cell height (<10% of a typical cell height, and ~1% of the embryo diameter), could result in small but systematic bias of the SOI around the cellularization front and impair projection quality. We avoid this problem by determining a new point cloud for each time point, for which we focus the ilastik detector on the interface between basal myosin and yolk. Our model approximates the embryo as a thin shell (see below), and hence as 2D surface. Therefore, we map the dynamic basal cell surface onto the cylinder grid of the static apical SOI.

## Particle image velocimetry

We measure the flow field using the particle image velocimetry (PIV) method, that identifies local displacements between two time points (*Adrian, 2005*). Briefly, we implemented the phase correlation method that leverages favorable execution times of fast fourier transforms, to estimate local flow in a region of interest on the projections (*Kuglin, 1975*). To minimize effect from systematic distortions towards polar regions on cylinder projections, we adjust the size of the ROI according to the local metric strain, which we define as the deflation of the metric from flat space (*Heemskerk and Streichan, 2015*).

## Reproducibility of the morphogenetic flow

Although gastrulation in *Drosophila* is highly reproducible from embryo to embryo (*Irvine and Wieschaus, 1994*), in practical terms experiments are subject to a constant time shift, depending on the developmental stage of the sample at the start of imaging. Thus, we developed an automated routine that allowed us to identify a common time frame that the 36 (WT: N = 22, *twist*:7, *bcd nos tsl*:7) live-imaged embryos are registered to. Specifically, we introduced a constant time shift for a given flow field that minimizes the squared difference with respect to a reference flow field averaged over the embryo surface:

$$\min_{t_{off,i}} \int < \sqrt{\left( \vec{v}_{ref}(t) - \vec{v}_i\left(t - t_{off,i}\right) \right)^2} >_{embryo} dt$$

where $<>_{embryo}$ denotes averaging across the embryo, $\vec{v}_{ref}$ is an arbitrary chosen reference from the ensemble, and $\vec{v}_i$, $t_{off,i}$ denote the i-th flow field and offset time respectively. In this way, we align samples to a chosen reference, in which we use the first occurrence of the cephalic furrow (CF) as a landmark indicating our choice for $t = 0$ min. Within a given genotype, we could automatically determine the offset time. However, to align mutants to WT, we first aligned all mutant datasets, and then used landmarks such as the CF (*twist*), or the VF (*bcd nos tsl*), to define a common time frame as best as possible.

Time shifted accordingly, we created an ensemble average flow field for each genotype:

$$<\vec{v}>_{ensemble} := \frac{1}{N} \sum_{i}^{N} \vec{v}_i\left(t - t_{off,i}\right)$$

The magnitude of ensemble average is highly reproducible from embryo to embryo (note the small standard deviation *Figure 1—figure supplement 5a*). Flow trajectories during cellularization

point towards the dorsal side (*Figure 1—figure supplement 5b*), showing persistent movement towards dorsal regions during pre-CF flow. This is accompanied by reduction of apical cell area in these regions (*Figure 1—figure supplement 5c*), as measured using confocal microscopy. While the length of pre-VF flow lines peaks on the anterior and posterior poles in WT, it is substantially reduced near poles in *twist*, and only mildly reduced in *bcd nos tsl* (*Figure 1—figure supplement 5d–f*). Together with the loss of basal DV asymmetry, this suggests that *twist* mediated reduction of basal myosin levels on the ventral side is responsible for dorsal-ward flow.

## Myosin quantification

### Intensity normalization

Using the imaging and pre-processing procedure as outlined above with samples expressing sqhGFP (*Royou et al., 2002*), we created projections of the apical and basal cell surfaces, with the goal of establishing a quantitative measurement of global myosin patterns (*Figure 1—figure supplements 6* and *7*). Ideally, quantification of signal intensities is carried out using identical conditions for each sample in the pool used for statistics, to minimize variability across samples. However, when performing *in toto* live-imaging, it is difficult to image more than one sample at a time and keep a high recording frequency. To minimize variability in a sequential recording scheme, we keep imaging conditions constant, but there are still possible variabilities in recorded signal intensity for biological but also technical reasons.

To account for such variability between experiments, we normalize recorded data (*Figure 1—figure supplement 6b,b',c*). Signal intensity of all time points in a given experiment are summarized by normalizing the intensity distribution: upper and lower range are determined according to the $ll = 0$ and $ul = 95$-percentile; normalization is done by subtracting the $ll$ and dividing by $(ul - ll)$, yielding a dimensionless normalized signal distribution (compare *Figure 1—figure supplement 6b and b'*). This strategy should not only allow for comparison on the same microscope, but also across microscopes, allowing for validation of *in toto* live-imaging from sequential experiments against fixed batches imaged e.g. on a confocal.

### Basal myosin pool analysis via light sheet microscopy

Here, we briefly outline the results for the DV asymmetry in the basal myosin pool that we reported in the main text. First, we time align intensity normalized basal projections as described above. Next we convolve each pullback with a Gaussian of width $\sigma \sim 3$ cell diameters to obtain basal myosin at the mesoscale (see discussion in model section below for definition). The results are then ensemble averaged to obtain ensemble myosin distribution as shown in *Figure 1—figure supplement 6*. To assess DV asymmetry, we focus on the region outlined by the black/white dashed line, where we first take an average along the AP axis and then compute average signal on dorsal side, and subtract from it the average signal on the ventral side. Repeating the outlined routine for all time points, we obtain the plot show in main text *Figure 4b*.

### Basal myosin pool via confocal microscopy

*Figure 1—figure supplement 7c* shows DV cross sections of fixed embryos stained for rb anti zipper and mouse anti dorsal, cut along the AP axis, and imaged on a confocal microscope. DV orientation of the samples is automatically determined based on the dorsal signal. To estimate the age of fixed embryos in relation to live-imaging data, we constructed a calibration curve for cell apico-basal height shown in *Figure 1—figure supplement 7a*. Using the known monotonic relation between cell height and age (*Merrill et al., 1988*), which we find lasts until about $8\ min$ after CF formation, we obtain estimate for the age of a fixed embryo based on measuring cell height. By segmenting the outline of basal myosin, we can then measure DV asymmetry in the same way as described for live-imaging data above.

A direct comparison between live imaging-based DV asymmetry measurement, and based on $N = 345$ fixed DV cross sections from confocal shows that after applying normalization routine as described above, we find similar estimates for the DV asymmetry using both light sheet and confocal imaging (see *Figure 1—figure supplement 7b*). Note that uncertainties in the calibration curve propagate to exact age determination in the embryo, and thus increased fluctuations in DV asymmetry determined using confocal imaging.

## Finite element implementation

Inversion of the continuum equation of state relating the coarse-grained myosin tensor and cellular flow-field was achieved using Finite Element Methods (FEM) in the weak formulation implemented within the FELICITY toolbox for MATLAB (*Walker, 2017*). Equations were inverted on a static triangular mesh representing the 'canonical' embryo surface produced via a point cloud (described above) subsequently turned into a smooth triangulation using MeshLab (*Cignoni, 2008*). As such, all objects within the equation of state had to be parameterized within the 3D embedding space of the mesh, which can be done by using the direction relation between projections and SOI (*Heemskerk and Streichan, 2015*). The only dynamic input to this inversion algorithm is the divergence of the myosin tensor. This was computed by interpolating the gradient of each Cartesian component of the tensor onto triangular faces of our mesh, producing a $3 \times 3 \times 3$ object on each face. The partial trace of this object over directions within the tangent plane of the face result in the estimated divergence. This operation was repeated for all myosin pools used. All equations within the FEM software are projected onto the surface of our 3D mesh to manually ensure solutions only exist within the tangent plane.

To benchmark the quality of our FEM solver, we implemented the equation of motion on a sphere and tested results for known solutions, which confirmed our solver works within the expected numerical accuracy. To test dependence on discretization used, we compared our results using different meshes at varying mesh sizes, and found good agreement with all test cases.

In order to model internalization of the VF, we introduced the ability to add a 'cut' within the triangular mesh. Contraction of the tissue was modeled by manually introducing local force dipoles on edges within the mesh pointing along the bond. All vertices along the cut were given zero bulk modulus to allow for local tissue compression needed to simulate invagination. The location of the cut was estimated from the PIV flow fields. Specifically the ratio of the divergence of the velocity field to the velocity field's magnitude was used to estimate the spatial extent of the cut over times during ventral furrow formation.

Predictions for the flow field obtained via inversion are subject to an overall scale factor, that can't be determined by the model. To compare ensemble averaged flow field measurement $\vec{v}(t)$ to model predictions $\vec{u}(t)$ in a quantitative fashion, we define a global measure for the spatial residual that is insensitive to such a scale factor. With the short hand notation $<\vec{u}> := \sqrt{<\vec{u}(x)^2>_{embryo}}$ to define overall magnitude of the field u across the surface of the embryo ($<\vec{u}(x)^2>_{embryo}$ denotes averaging the space dependent field $\vec{u}(x)^2$ across the embryo surface, so is not space dependent.), the residual is defined as

$$R = \frac{\left(<\vec{u}>^2\vec{v}(x)^2 + \vec{u}(x)^2<\vec{v}>^2\right) - 2\sqrt{<\vec{u}>^2<\vec{v}>^2}\vec{v}(x)\vec{u}(x)}{2<\vec{u}>^2<\vec{v}>^2}$$

provides a spatial discrepancy map, indicating the prediction quality as a function of location on the embryo, that is in-sensitive to noise dominated fluctuations in domains of no flow (i.e. fixed points), as opposed to e.g. inner product.

## Acknowledgements

We thank Trudi Schüpbach as well as members of Shraiman and Wieschaus labs for helpful discussions; Lars Hufnagel for initial support with the MuVi SPIM setup; and Reba Samanta for preparing histological samples used to create *Figure 1—figure supplement 7*. This work was funded by the Gordon and Betty Moore Foundation grant #2919 (SJS), NSF PHY-1220616 (BIS), grant #1K99HD088708-01 from National Institute of Child Health and Human Development (SJS). EFW is an investigator with the Howard Hughes Medical Institute.

## Additional information

### Funding

| Funder | Grant reference number | Author |
| --- | --- | --- |
| National Science Foundation | PHY-1220616 | Boris I Shraiman |
| Howard Hughes Medical Institute | | Eric F Wieschaus |
| National Institutes of Health | NICHD 1K99HD088708 | Sebastian J Streichan |
| Gordon and Betty Moore Foundation | GBMF #2919 | Boris I Shraiman |

The funders had no role in study design, data collection and interpretation, or the decision to submit the work for publication.

### Author contributions

Sebastian J Streichan, Conceptualization, Resources, Data curation, Software, Funding acquisition, Investigation, Visualization, Methodology, Writing—original draft, Writing—review and editing; Matthew F Lefebvre, Resources, Investigation, Writing—review and editing; Nicholas Noll, Conceptualization, Software, Formal analysis, Investigation, Writing—review and editing; Eric F Wieschaus, Conceptualization, Resources, Funding acquisition, Investigation, Methodology, Writing—review and editing; Boris I Shraiman, Conceptualization, Supervision, Funding acquisition, Investigation, Methodology, Writing—original draft, Writing—review and editing

### Author ORCIDs

Nicholas Noll https://orcid.org/0000-0003-1698-7500
Eric F Wieschaus https://orcid.org/0000-0002-0727-3349
Boris I Shraiman http://orcid.org/0000-0003-0886-8990

### Decision letter and Author response

Decision letter https://doi.org/10.7554/eLife.27454.023
Author response https://doi.org/10.7554/eLife.27454.024

## Additional files

### Supplementary files

• Transparent reporting form
DOI: https://doi.org/10.7554/eLife.27454.019

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

## Appendix 1

DOI: https://doi.org/10.7554/eLife.27454.020

# Segmentation free anisotropy detection using Radon transforms

When viewed in cross-section, the cell cortex in epithelia forms a polygon tiling, where the outline of each cell is well approximated by a closed sequence of edges. Anisotropic distribution of proteins is often characterized by homogeneously increased accumulation to cell edges of particular orientation, while it remains homogeneously low and comparable to background on other edges (*Zallen and Wieschaus, 2004*; *Bertet et al., 2004*; *Blankenship et al., 2006*). Note that typically the number of edges at low and high signal accumulation is of the same order of magnitude. Available methods to quantify cortical anisotropy mostly operate at the single cell level and construct a nematic tensor by integrating signal intensities along cell outlines (*Aigouy et al., 2010*). At the organismal scale membrane segmentation is costly, and for polarized markers low signal to noise on a significant number of edges often results in difficulties to close the cell circumference. We overcome the need for fiduciary markers that increase experiment complexity, by shifting perspective to cell edges and designing a robust and rapid segmentation free anisotropy detection algorithm.

Let's consider an image on a rectangular domain $\Omega$, showing an edge of possibly non-uniform intensity, assigning each pixel at coordinates $(x, y) \in \mathbb{R}^2$ some intensity $I(x, y) \in \mathbb{R}$ - as shown e.g. in *Figure 2—figure supplement 1*. Lets call this edge a **linear signal** or just signal. For simplicity, and without loss of generality, we assume the average intensity along the linear signal is $<I_{ls}> = 1$, and average background intensity is $<I_{bg}> = 0$. While under favorable conditions with high signal to noise ratio, it may be possible to identify the linear signal using conventional methods such as edge detection using a typical threshold, this task will be substantially more error prone at low signal to noise ratio.

Therefore, we decided to reformulate the problem using Radon transforms (*Radon, 1917*), that integrate signal along lines of a given angle and signed distance from the origin. Consider a line $L$ with normal $\vec{n}_L = (\cos\alpha, \sin\alpha)$, a signed normal distance $\delta \in \{-\infty, \infty\}$ away from the origin (e.g. shown in pink *Figure 2—figure supplement 1*, top left), which may be parameterized as follows

$$\begin{pmatrix} x(t) \\ y(t) \end{pmatrix} = \begin{pmatrix} t\sin\alpha + \delta\cos\alpha \\ -t\cos\alpha + \delta\sin\alpha \end{pmatrix}$$

where $t \in \{0,1\}$ is the parameter that marks position along the line $L$ (*Figure 2—figure supplement 1*). It is clear that all possible pairs of normal orientation and signed distance $(\alpha, \delta)$, describes all possible lines covering the plane. The **Radon transform** of the image $I$ - denoted $RI$ - establishes a map from the rectangular domain $\Omega$, on which the image is defined into the space spanned by line orientation and signed distance, where each line $L$ characterized by the pair $(\alpha, \delta)$ is assigned the integrated(summed) intensity projection along the line - or in mathematical terms:

$$RI(L) = \int_L I(x, y) d\Omega$$

$$RI(\alpha, \delta) = \int_0^1 I(t\sin\alpha + \delta\cos\alpha, -t\cos\alpha + \delta\sin\alpha) d\Omega$$

where the latter equation uses the explicit parametrization of the line $L$ in terms of $\alpha, \delta, t$ defined above. (The following is a verbal example for the equation above, and may be skipped): *Figure 2—figure supplement 1* illustrates the algorithm that constructs the radon transform for the linear signal shown in white, one that has a normal orientation of $45^0$ and constant intensity $I_{ls} = 1$ over a background of intensity $I_{bg} = 1$. The magenta dashed line

indicates a line $L$ with normal orientation $\alpha$ and a distance $\delta = 0$ away from the origin shown in red. Integrating image intensity along such a line, for any orientation $\alpha \neq 45^0$, we theoretically obtain only a contribution from the intersection with the linear signal, thus $RI(\alpha \neq 45^0, 0) = 1$. In contrast for $\alpha = 45^0$, i.e. when the line is parallel and on top of the linear signal, the radon transform integrates the intensity along the entire line, returning the length $l$ of the linear signal: $RI(\alpha = 45^0, 0) = l$. In our example, for $\delta \neq 0$ we will either have a single intersection of the line $L$ with the signal, again returning with the same outcome as above, or no intersection, where the radon transform returns 0, indicated by the dark blue regions in *Figure 2—figure supplement 1*.

Since the radon transform is linear, it follows that for any image that is the linear superposition of linear signals, the radon transform is the linear superposition of the radon transform of each linear signal. For example, if an image consisted of 2 linear signals (for example a four fold vertex), the resulting radon transform is the sum of the radon transform of each linear signal, with peak levels at orientation and signed distance of each linear signal. In this way, linear signals are mapped to peaks in the radon transform that reflect the total intensity along the length of the linear signal. Knowing the location of peaks in the radon-transformed space allows us to reconstruct the position and orientation of each linear signal in the image. This simplifies the task of identifying the linear signal, as detection of peaks due to their compact structure is simpler than detection of edges, and the radon transformed signal will be less susceptible to fluctuations in the underlying data, and therefore enhance robustness.

To obtain average image intensity from the radon transform we use the fact that the height of the peak reflects the total intensity $I_{tot}$ along the linear signal to construct the average signal as $<I> = I_{tot}/l$, where $l$ denotes the length of the linear signal. To determine the length of the linear signal, we need to determine the position where the linear signal begins and/or ends in the image. This is done by interpolating the signal along the orientation and position corresponding to the identified peak in the radon transform, and identifying discontinuities in the interpolated signal, which mark the boundaries of the linear signal. With beginning and end determined, we obtain the length as the magnitude of the vector connecting the two points. If beginning and end don't fall onto the boundary of the image, we perform an integration of a line restricted between these two endpoints, to obtain a more accurate estimate of the total line intensity.

## Validation of segmentation free anisotropy detection

Pullbacks created with ImSAnE based on light sheet microscopy data of developing *D. melanogaster* embryos show the cortex of roughly 6000 cells, which would yield a complex radon transform pattern, effectively impeding peak detection. To simplify peak detection, we decided to carry out analysis in small blocks, where we focus on a local region of interest (ROI) that we sweep across the entire image. The size of the local region is chosen such that it can accommodate between 1–3 typical edge lengths $l_{edge}$. In this way, we typically obtain a small number (<10) of edges in the ROI, such that we could use the extended-maxima transform (*Soille, 2013*) to identify the typically well separated peaks. To avoid double counting of edges - as they may appear from sweeping the ROI - we average detected lines of locally similar angles with a small angular difference $\|\delta_\alpha\| < \epsilon$ and distance $\delta < l_{edge}$.

Following the general prescription given above, we obtain local estimates for position, orientation, and average intensities of cell edges, demonstrated in *Figure 2—figure supplement 2*. A zoom on a local region indicated by the white box shows that this routine - fully automatically, and without need of fine tuning parameters - faithfully detects the orientation of cell edges, including dim edges with only minimally stronger signal compared to background. To further benchmark the quality of intensity estimates, we turned to known anisotropy of myosin during germband extension (*Blankenship et al., 2006*). *Figure 2—figure supplement 1* shows on the right, that the presented algorithm finds the normalized intensity on AP edges ($90^0$) is systematically higher than on DV edges ($0^0$), reaching is maximum

approximately 20 min after cephallic furrow formation, in good agreement with previous manual measurements (*Blankenship et al., 2006*).

# Appendix 2

DOI: https://doi.org/10.7554/eLife.27454.021

## Mathematical model of tissue flow

Mechanics of epithelial tissue is largely defined by the properties of intracellular cytoskeletal cortexes, linked by cadherin mediated adherens junctions into a global trans-cellular mechanical network. This mechanical network is 'active' in the sense that it involves myosin motors that generate internal forces and can do work by contracting actin-myosin filaments under tension. A cytoskeletal network is also 'adaptive' in the sense that it can relax mechanical stress by reorganizing internally, on sub-cellular scale by recruiting (or releasing) myosin and other key molecular components, and on tissue scale, by allowing cells to rearrange locally (by T1 processes). The latter process allows cells to change neighbors and 'flow' relative to each other, while preserving the integrity of the epithelium and its cytoskeletal network. The detailed description of these complex cellular processes is not necessary for our present goals, which call merely for an approximate mesoscale description of tissue mechanics. By mesoscale we mean the scale of >3 cell diameters, still much smaller than the macroscopic scale of the embryo tissue flow. Microscopic complexity notwithstanding, on mesoscale we will think of tissue as a continuous medium and it will suffice for us to capture the facts that i) on short time scales tissue responds elastically to mechanical perturbations (**Bambardekar et al., 2015**) and ii) on the longer time scales elastic stress is relaxed through active rearrangement of the cytoskeleton as cells adapt to the imposed deformation. This is enough to define flow velocity in response to external and internal stress: on the timescale comparable to internal stress relaxation, tissue dynamics can be described by a generic viscoelasticity equation with two effective viscosity parameters which we shall now derive.

Short-term elastic response means that incremental increase of strain causes an increase in stress, which can subsequently relax with the relaxation time $\tau_R$. To simplify the derivation we will present now, lets assume a flat two-dimensional surface. Elastic stress is then governed by the Maxwell viscoelasticity and in the mesoscopic continuum approximation we have:

$$\dot{\sigma}_{ab} = \mu(\partial_a \dot{u}_b + \partial_b \dot{u}_a) + \lambda \delta_{ab} \partial_c \dot{u}_c - \tau_R^{-1} \sigma_{ab} \tag{1}$$

where $\dot{x} = \frac{d}{dt}x$, so that $\dot{u}_a$ is the rate of local displacement in the direction of spatial component a. Spatial derivatives of the $\dot{u}_a$ vector define local rate of strain $\partial_a \dot{u}_b + \partial_b \dot{u}_a$. In addition, $\delta_{ab}$ is the Kronecker delta matrix and we have adopted the Einstein convention of summing over repeated indices (e.g. $\partial_c u_c = \sum_c \partial_c u_c$). The first two terms on the right hand side describe generation of stress in proportion to the rate of strain ($\mu$, $\lambda$ are the Lame coefficients parameterizing an elastic stress-strain relation) and the last term parameterizes relaxation of stress.

On the other hand, in the continuum approximation, tissue flow velocity $v_a = \dot{u}_a$ can be described by the (compressible) Stokes equation

$$\rho \dot{v}_a = \upsilon_0 \partial_b^2 v_a + \partial_b \sigma_{ab} + F_a \tag{2}$$

where $F_a$ is external force (per unit area), $\rho$ is the density and $\upsilon_0$-dynamic viscosity.

Now suppose that stress relaxation is sufficiently rapid to achieve quasi-equilibrium

$$\tau_R^{-1} \sigma_{ab} \approx \mu(\partial_a v_b + \partial_b v_a) + \lambda \delta_{ab} \partial_c v_c \tag{3}$$

substituting $\sigma_{ab}$ into **Equation 2** we have

$$\rho \dot{v}_a = (\upsilon_0 + \tau_R \mu)\partial_b^2 v_a + \tau_R(\mu + \lambda)\partial_a \partial_b v_b + F_a \tag{4}$$

Note that transient elasticity, parameterized by $\tau_R$, $\mu$ and $\lambda$, generates 'effective viscosity' $\tau_R \mu$, which can dominate the microscopic viscosity $\upsilon_0$ of the 'fluid' itself. This effective viscosity is in general anisotropic with $\tau_R \mu$ contributing to the shear viscosity which resists shear, acting

to make flow more uniform, and the bulk viscosity $\tau_R(\mu + \lambda)$ which acts specifically on the on the compressible component of the flow.

If the effective viscosity is sufficiently high and flow changes sufficiently slowly, the inertial term can be neglected ($\rho\dot{v}_a \ll v\partial_b^2 v_a$) and the quasi-stationary flow is defined by the balance of the external forcing and effective viscosity

$$v_1\partial_b^2 v_a + v_2\partial_a\partial_b v_b = -F_a \tag{5}$$

where $v_1 = (v_0 + \tau_R\mu)$ and $v_2 = \tau_R(\mu + \lambda)$. Since external force is related to external stress via $F_a = \partial_b\sigma_{ab}^{ext}$ and we have argued that relevant 'external stress' that drives tissue flow in the embryo is proportional to the myosin tensor $\sigma_{ab}^{ext} \sim m_{ab}$ we have arrived at the equation we have used in the main text to relate tissue flow velocity with myosin distribution. (Note that the relation of $\sigma_{ab}^{ext}$ and $m_{ab}$ is due to the fact that myosin generates contractile forces, so that a local maximum of an isotropic myosin distribution $m_{ab}(r) = \delta_{ab}m(r)$ would act just like a low pressure region, generating inward directed flow.)

$$v_1\partial_b^2 v_a + v_2\partial_a\partial_b v_b = -\partial_b m_{ab} \tag{6}$$

## Passive versus active mechanics

In the section above we put forward a simple effect model based on conventional Maxwell viscoelasticity. Epithelial tissues however are clearly not ordinary passive viscoelastic materials. Myosin-driven rearrangement of the cytoskeleton is an active and adaptive process, which can be regulated on cellular and subcellular scales. We argue, however, that average flow on the mesoscopic scale can be usefully approximated by the passive viscoelastic model with suitable effective parameters. The ability of this simple model to capture highly non-trivial spatial flow patterns without the need for introducing spatial dependence of model parameters proves the validity of the approximation. Still, we do not expect this 'passive viscoelasticity' model to be complete. We anticipate that a more detailed description of myosin activity and myosin recruitment dynamics would be required in order to describe both the fast cellular scale fluctuations and the long-term myosin dynamics on the scale of the embryo. Other molecular/genetic factors will come into play as well: e.g. cadherin and other cytoskeletal components on small scale and transcription factors that guide morphogenesis on large scales. We plan to address these issues in the future.

