## [Decision Letter]

Thank you for submitting your article "Global morphogenetic flow is accurately predicted by the spatial distribution of myosin motors" for consideration by *eLife*. Your article has been reviewed by three peer reviewers, including a member of our Board of Reviewing Editors, and the evaluation has been overseen by the Reviewing Editor and Anna Akhmanova as the Senior Editor.

The reviewers have discussed the reviews with one another and the Reviewing Editor has drafted this decision to help you prepare a revised submission.

Summary:

The authors perform a careful and thorough analysis of the collective cell movements during early development of the *Drosophila* embryo. Using light sheet microscopy and a quantitative image analysis the observed patterns of fluorescence intensity are mapped on a closed two-dimensional surface representing the epithelial surface. In the case of fluorescently labelled myosin, patterns of myosin on the apical and on the basal side of the tissue have been quantified separately on the 2d surface. The work uses an elegant and original approach to quantify anisotropies of apical myosin using a Radon transform. In addition, flow fields are determined using PIV. The authors use the 2d distribution and anisotropies of apical and basal myosin to estimate patterns of active stress in the tissue. Using a continuum theory of an active gel, velocity fields are calculated and compared to the observed cell flow patterns. Strikingly, the authors show that the most important features of the cell flow can be accounted for from the knowledge of myosin distributions. Furthermore, the authors show that altered flow patterns in *twi* mutants can be explained by changed patterns of myosin distributions under mutant conditions. These are important results that provide a significant advance of the understanding of morphogenetic movements in the early fly embryo. The quantitative methods used are powerful and sophisticated. While this work is interesting and strong, there are several points that need to be addressed in a revision of the manuscript and several points need clarification.

Essential revisions:

1) A major concern as detailed below is that a lot of information is missing in the manuscript.

2) The authors do not show what the actual myosin distribution looks like, and there is little detail in how the 'coarse-grained' or 'smoothened' myosin distributions in Figure 1 are obtained. The corresponding supplemental figure did not really help in this regard. If the data has been smoothened to produce these plots, the original data should also be shown. Scale bars should also be added to the figures. What is the 'contrast' in the myosin signals? What is the level of intensity between dorsal and ventral in relation to the average? What are the units in Figure 1, Figure 2, etc.?

3) The model used to calculate the flow fields is shown in Figure 3. It is unclear why the isotropic basal and the anisotropic apical signals are used only. The apical myosin also has an isotropic signal. Is this not used and why not? Maybe the notation, which is not fully explained, leads to confusion here. Is the tensor m_apical_ traceless or does it contain an isotropic signal? This should be clarified. Presumably the isotropic and anisotropic contributions of apical myosin are contributing with two different factors. These factors should be defined and their values reported.

4) The quantification of anisotropies of the myosin distribution represents an interesting methodological advance. The presentation of the quantification of this anisotropy on the surface of the embryo is not very clear. Figure 2 and Figure 2 do not provide a full picture of the quantified anisotropy patterns. The authors should show a complete map of myosin anisotropy at different times.

5) Information about the anisotropy of the basal signal evaluated with the same method is missing. Why is the basal myosin anisotropy not contributing to the stress equation? Is it because basal myosin anisotropy is low?

6) In the appendix the anisotropy quantification using a Radon transform is explained. However, I did not see a clear definition of how the Radon transform information is used to determine the myosin tensor m_apical_ in Figure 3.

7) If m(r) denotes a factor positively correlated to myosin concentration, the sign of the righthand side of Equation 6 may be wrong. With the current sign, a positive accumulation of myosin would drive an outward flow. It is essential that the authors clarify this.

8) In Figure 2 it is not clear what is plotted. The caption states "Eigenvectors of myosin tensor are plotted in cyan". It is not clear if the myosin tensor is a field defined on a regular grid, of only selected vectors are shown. Shown are bars that are hardly visible to the bare eye. They do not look like vectors and it is unclear to which points they are attached.

9) The authors should provide more information on the 2D shear and bulk viscosities ν and ν' and they ratio B. If one considers the height of the tissue layer as an explicit variable and treats the tissue as isotropic and incompressible in 3D, divergences in the 2D flow field couple to height changes and the ratio of the 2D bulk to the 2D shear viscosity would be 3 (see e.g. Batchelor, 2000). How does this compare to the values that are fitted here? Can the model be simplified to 2 parameters? Can B be constant in time (I assume not, Figure 1)?

10) Related comment: Please provide the values of the parameters B, α, β in a table/plot in the main text together with confidence bounds. Are α and β constant in time? The 'true' number of model parameters remains unclear. Is this really just one parameter as stated in the abstract, or is it three as stated in the main text? It remains unclear whether the viscosity ratio B was different for each time point which would correspond to effectively many fit parameters. In the main text it is mentioned that B changes with time however this is unclear and needs to be clarified. The main text refers to Figure 1 about this but in the figure there is no information about B.

11) It seems the authors use a single value of B that is space-independent at each time point, but then allow B to instantaneously change everywhere and synchronously in space. The motivation for this remains unclear. How could that be achieved in the embryo?

12) In Figure 1 it is not stated in the caption (and not even in the supplement) which fluorescence signal (apical, basal myosin, combined?) is used to calculate the velocity field. This important information that should be found easily.

13) In Figure 3, it is unclear whether the shown flow profiles are obtained with or without the "cut" in mesoderm? We would suggest adding a figure or schematic actually showing what the "cut in mesoderm" perturbation is doing.

14) One sentence states that the comparison between theoretical and experimental plots is done using normalised velocity fields rather than real velocity fields (caption of Figure 3: "Fit residual, comparing predicted flow field with measured flow field normalized for magnitude"). The precise way the velocities were normalized remained unclear to the referees. Was the velocity normalized at each position separately or was only the overall magnitude of the velocity field normalized? The former case would be a problem because real velocity fields should be compared including velocity amplitude patters.

15) An important result is that "the model achieves about 90% accurate description of the flow pattern before and after VF invagination". It is quite unclear how this percentage of agreement is defined and determined. Also, in the caption of Figure 3 the residual is not defined. The definition of the residual given in the methods remains a bit unclear because the notation is not well explained (which quantities depend on position and which do not, how is <[…]>_embryo_ defined etc.).

16) The flow and myosin profiles in twist embryos are not reported. They should absolutely be shown.

17) It is not clear what information the theoretical analysis of mutants is bringing. Are the parameters used to fit the data the same as in WT?

18) It is misleading to refer to the model used in the calculations as viscoelastic. This is a description in the fluid limit and it should be referred to as such. Viscoelasticity does not enter; the corresponding viscoelastic relaxation time is not a parameter that is considered. Please correct corresponding references in the Abstract and main text, for example the language in the Results section (please add page and maybe line numbers in the revision) is confusing and the statement in the Abstract is misleading (surprisingly simple effective visco-elasticity model -> viscous model). Of course, one can arrive at a fluid-like description by considering the long-time behavior of a viscoelastic material as the authors also show, and it is certainly ok to provide this derivation. But the theory that is used in the end is fluid and not being clear here can put the reader on a wrong track.

19) In Figure 3 a coordinate independent, covariant expression of the dynamic equations is presented. The equations given in the appendix are not covariant but depend on a coordinate system. It is therefore unclear whether the continuum theory applied to the problem is covariant or not. This should be clarified. Furthermore, the finite element method used is also not clear with respect to this point and not sufficiently explained. Do the results depend on the choice of the finite element discretization used or not? If not, has this been checked? If yes, what does this mean? We understand that the authors do not want to go into notational complexities here but the conceptual approach used should be clear.

---

## [Author Response]

Essential revisions:1) A major concern as detailed below is that a lot of information is missing in the manuscript.

We thank the referees for the careful reading of the manuscript and suggested improvements. Below, we discuss the individual points of concern which have been addressed largely through modified, and additional Supplementary figures.

The main changes are:

Figure 1—figure supplement 1–4,

Old Figure 1—figure supplement 1–3 are now Figure 1—figure supplement 5–7,

Figure 2—figure supplement 3,

Figure 2—figure supplement 4,

Figure 3—figure supplement 1,

Figure 4—figure supplement 1.

2) The authors do not show what the actual myosin distribution looks like, and there is little detail in how the 'coarse-grained' or 'smoothened' myosin distributions in Figure 1 are obtained. The corresponding supplemental figure did not really help in this regard. If the data has been smoothened to produce these plots, the original data should also be shown. Scale bars should also be added to the figures. What is the 'contrast' in the myosin signals? What is the level of intensity between dorsal and ventral in relation to the average? What are the units in Figure 1, Figure 2, etc.?

We have added Figure 1—figure supplement 1, to demonstrate the initial steps of the imaging data analysis pipeline, consisting of defining embryo shape, apical, basal, and lateral surfaces. We explicitly define the “midplane” section, cutting across the lateral surfaces of cells, that was used as a common reference frame, and the apical and basal surfaces related by normal displacement. The individual surfaces are color-coded and original signal on each surface shown. The midplane section surface was used to evaluate the flow field using the PIV method (as described in the Materials and methods section), which we indicated by adding a flow field next to it. To show the time course of myosin signal on apical and basal surfaces respectively, we added Figure 1—figure supplement 2, Figure 1—figure supplement 3 and Figure 1—figure supplement 4. We also provide measures for contrast, defined as root mean square contrast, shown in Figure 1—figure supplement 2.

Original myosin signal is measured in arbitrary units. In the Material and methods section, we describe how we turn this into a dimensionless number by normalization. Therefore, all plots showing the myosin tensor have no units.

To better visualize the normalization we added panels B and B’ to Figure 1—figure supplement 2. Also, we added panel C, to report requested statistics on dorsal and lateral regions respectively.

We added scale-bars to all projections of original embryo surfaces.”Pullbacks” (onto the cylinder) are locally distorted and thus a single scalebar would be misleading.

To explain the coarse-graining procedure, we added Figure 2—figure supplement 3, which in panel (A) shows outlines of cells, and the coarse-grained lattice. In panel (B) we show how the myosin tensor is defined on the mesoscale, which involves smoothing with a Gaussian kernel. All relevant quantities are indicated in the panel (see below comments).

3) The model used to calculate the flow fields is shown in Figure 3. It is unclear why the isotropic basal and the anisotropic apical signals are used only. The apical myosin also has an isotropic signal. Is this not used and why not? Maybe the notation, which is not fully explained, leads to confusion here. Is the tensor m_apical_ traceless or does it contain an isotropic signal? This should be clarified. Presumably the isotropic and anisotropic contributions of apical myosin are contributing with two different factors. These factors should be defined and their values reported.

We thank the refees for pointing out the potentially confusing notation. Now the annotation for the model shown in Figure 3 shows that the isotropic basal pool is represented by the scalar field ‘m_basal_’, (we explain below why we neglected anisotropic contributions). The apical pool is represented as the full tensor ‘m_apical_’, which, as we explain in the text, is not traceless and has anisotropic and isotropic contributions, as shown in Figure 3. To this end we explain in the annotation in Figure 3 how the tensor decomposes into an isotropic scalar (i.e. the trace), and an anisotropic traceless part. In this way there is no need for an additional factor distinguishing between isotropic and anisotropic contributions in the apical pool. The point is emphasized in the revised Results section:

“By averaging the resulting tensors in a given region, we obtain a quantitative description of local tissue anisotropy and overall levels that reflects the intensity-weighted average of cell edges. The resulting tensor generally has a trace, and thus can be separated into an isotropic and a traceless anisotropic part

(Figure 3).”

4) The quantification of anisotropies of the myosin distribution represents an interesting methodological advance. The presentation of the quantification of this anisotropy on the surface of the embryo is not very clear. Figure 2 and Figure 2 do not provide a full picture of the quantified anisotropy patterns. The authors should show a complete map of myosin anisotropy at different times.

We thank the referees for these suggestions and added Figure 2—figure supplement 3, where we explain how the coarse-grained myosin tensor on the mesoscale is obtained. Panel (B) emphasizes the distinction between isotropic and anisotropic parts, to provide a clear connection to the quantitative measure of anisotropy in basal and apical pools shown in panels (C and D). We further provide a sequence of representative time-points for the apical surface, showing that myosin anisotropy is strongest in the germband and peaks around 17 minutes. For better intuition, we also added plots on the apical surface of the embryo in panel (E). For a complete picture of Figure 2, we demonstrate a time series of the principal axes of the anisotropic myosin tensor.

5) Information about the anisotropy of the basal signal evaluated with the same method is missing. Why is the basal myosin anisotropy not contributing to the stress equation? Is it because basal myosin anisotropy is low?

As correctly guessed by the referees, we neglect basal pool anisotropy in our study, because it is small. We added Figure 2—figure supplement 3 to demonstrate this.

6) In the appendix the anisotropy quantification using a Radon transform is explained. However, I did not see a clear definition of how the Radon transform information is used to determine the myosin tensor m_apical in Figure 3.

Figure 2 outlines the concept of how we describe edge orientation r(i), based on the detected angle α using the radon transform. Below it we show how the myosin tensor m_ab_ on each edge is computed. To make the connection with the model clearer, we changed the notation, to follow the convention in Figure 3, and indicated that we compute the tensor for each edge (i) but only on the apical surface by specifically referring to m_apical(i)_.

To clarify how we compute the myosin tensor on the mesoscale, we added Figure 2—figure supplement 3, which demonstrates the conversion from nematic m(i) obtained by Radon transform on cell edges to a regular lattice covering the embryo surface. Panel (A) shows a sketch of a cell array in green, and points in the regular lattice covering the embryo in black. From this we compute distance between a given edge i, and lattice site x, which is used to construct a Gaussian weight in the definition of the mesoscale myosin tensor, explained in panel (B).

7) If m(r) denotes a factor positively correlated to myosin concentration, the sign of the righthand side of Equation 6 may be wrong. With the current sign, a positive accumulation of myosin would drive an outward flow. It is essential that the authors clarify this.

Of course, the referee is correct, in that a positive accumulation of myosin should create an inward flow. We have corrected the misprint in the equation.

8) In Figure 2 it is not clear what is plotted. The caption states "Eigenvectors of myosin tensor are plotted in cyan.". It is not clear if the myosin tensor is a field defined on a regular grid, of only selected vectors are shown. Shown are bars that are hardly visible to the bare eye. They do not look like vectors and it is unclear to which points they are attached.

We added Figure 2—figure supplement 3 that explains how the myosin tensor field is defined. In the caption to Figure 2 we now explain that we show the axis parallel to the leading eigen-vectors of the tensor field, which we only plot along even skipped stripes for simplicity of comparison.

9) The authors should provide more information on the 2D shear and bulk viscosities ν and ν' and they ratio B. If one considers the height of the tissue layer as an explicit variable and treats the tissue as isotropic and incompressible in 3D, divergences in the 2D flow field couple to height changes and the ratio of the 2D bulk to the 2D shear viscosity would be 3 (see e.g. Batchelor, 2000). How does this compare to the values that are fitted here? Can the model be simplified to 2 parameters? Can B be constant in time (I assume not, Figure 1)?

We have reworded the Discussion section in the text to clarify that viscosities in question are “effective” quantities that relate to the shear and bulk moduli and the rate of stress relaxation (Figure 3, more fully explained in Appendix 2) rather than derived from the isotropic and incompressible 3D fluid as suggested by the referee. As explained in the response to comment 10 and 11 below, B(t) (we updated the notation, to reflect which variables depend on space or time) is time dependent. However, it is low through much of cellularization and only increases significantly from the onset of cephalic furrow formation (T= 0 min) (see Figure 3—figure supplement 1). Cells already completed much of their elongation before (compare Figure 1—figure supplement 3). As outlined in the response to comment 11 below, we instead speculate the steady increase of incompressibility relates to completion of cellularization, which occurs concomitant with the apparent increase of the bulk modulus.

10) Related comment: Please provide the values of the parameters B, α, β in a table/plot in the main text together with confidence bounds. Are α and β constant in time? The 'true' number of model parameters remains unclear. Is this really just one parameter as stated in the abstract, or is it three as stated in the main text? It remains unclear whether the viscosity ratio B was different for each time point which would correspond to effectively many fit parameters. In the main text it is mentioned that B changes with time however this is unclear and needs to be clarified. The main text refers to Figure 1 about this but in the figure there is no information about B.

We thank the referees for pointing out the inconsistency between our original Abstract and the main text. We corrected the statement that there is “only one parameter”, to now say there is” only one time dependent and two constant parameters”. Indeed, B is a number that changes with time. In the added Figure 3—figure supplement 1, we elaborate on the number of model parameters, and explain that they are not spatially dependent. We stress that the complex spatial structure of the flow at any given time is accurately fitted with only one parameter, as the time-independent constants are determined by other time-slices. We also show how B depends on time, and provide values including confidence intervals of the constant conversion factors α and β.

11) It seems the authors use a single value of B that is space-independent at each time point, but then allow B to instantaneously change everywhere and synchronously in space. The motivation for this remains unclear. How could that be achieved in the embryo?

Updated Figure 3 annotation now indicates that apical and basal myosin is a field, while B(t) is a space-independent but time dependent scalar.

To improve presentation of this aspect in our model, we added Figure 3—figure supplement 1. Panel (B) shows the smooth temporal dependence of B, starting out low, and increasing slowly with cellularization. After gastrulation is completed, B is constant high. In the text, we also added the following statement:

“The temporal coincidence between completion of cellularization and increase of the bulk modulus provides an intriguing possible explanation of how the continuous transition in our time dependent variable might be realized.”

12) In Figure 1 it is not stated in the caption (and not even in the supplement) which fluorescence signal (apical, basal myosin, combined?) is used to calculate the velocity field. This important information that should be found easily.

We have added a figure, Figure 1—figure supplement 1, explaining the imaging pipeline used to extract the flow field, shown in panel (I) and define apical and basal myosin distributions

13) In Figure 3, it is unclear whether the shown flow profiles are obtained with or without the "cut" in mesoderm? We would suggest adding a figure or schematic actually showing what the "cut in mesoderm" perturbation is doing.

Following this suggestion and added a schematic, showing the cut in the mesoderm in the new Figure 3—figure supplement 1. We also explain in the caption that flow profile shown in Figure 3 was produced using the ‘cut’.

14) One sentence states that the comparison between theoretical and experimental plots is done using normalised velocity fields rather than real velocity fields (caption of Figure 3: "Fit residual, comparing predicted flow field with measured flow field normalized for magnitude"). The precise way the velocities were normalized remained unclear to the referees. Was the velocity normalized at each position separately or was only the overall magnitude of the velocity field normalized? The former case would be a problem because real velocity fields should be compared including velocity amplitude patters.

The caption of Figure 3 now reads:

“Fit residual, comparing predicted flow field with measured flow field (see subsection “Finite element implementation” for a detailed definition of the residual) as a function of time. Both fields are normalized for average magnitude. The average magnitude of predicted velocity field defines one of our fitting parameters.”

We also clarified our notation in the residual definition, to avoid potential

confusion (see response to comment 15).

15) An important result is that "the model achieves about 90% accurate description of the flow pattern before and after VF invagination". It is quite unclear how this percentage of agreement is defined and determined. Also, in the caption of Figure 3 the residual is not defined. The definition of the residual given in the methods remains a bit unclear because the notation is not well explained (which quantities depend on position and which do not, how is <[...]>_embryo_ defined etc.).

The caption of Figure 3 points towards the Materials and methods section for more detail. To make this link clearer, we now emphasize the residual is defined in subsection “Finite element implementation”.

In subsection “Finite element implementation”, we improve notation interpretation, explain the averaging procedure across the embryo, and emphasize which quantities in the residual depend on space. The SI text now defines:

With the short hand notation <u→>:=<u→(x)2>embryo to define overall magnitude of the field u across the surface of the embryo <u→(x)2>embryo denotes averaging the space dependent field u⃑(x)2 across the embryo surface, so is not space dependent.), the residual is defined asR=(<u→>2v→(x)2+u→(x)2<v→>2)−2u→>2<v→>2v→(x)u→(x)2<u→>2<v→>2

16) The flow and myosin profiles in twist embryos are not reported. They should absolutely be shown.

We have added Figure 4—figure supplement 1 to demonstrate representative time points of the flow field, basal, apical isotropic, and apical anisotropic myosin in twist mutants.

17) It is not clear what information the theoretical analysis of mutants is bringing. Are the parameters used to fit the data the same as in WT?

We thank the referees for pointing out the missing description. The caption of Figure 4 now states that model parameters are as previously determined for WT, and we emphasize in the Discussion section:

“Interestingly in-silico perturbations indicate the local depletion (on the ventral side) has a global effect, most evidently manifested by a “sink” on the dorsal side, which is lost in a simulation using the same parameters as WT, but no DV modulation of the basal myosin pool (Figure 4, left panels).”

18) It is misleading to refer to the model used in the calculations as viscoelastic. This is a description in the fluid limit and it should be referred to as such. Viscoelasticity does not enter; the corresponding viscoelastic relaxation time is not a parameter that is considered. Please correct corresponding references in the Abstract and main text, for example the language in the Results section (please add page and maybe line numbers in the revision) is confusing and the statement in the Abstract is misleading (surprisingly simple effective visco-elasticity model -> viscous model). Of course, one can arrive at a fluid-like description by considering the long-time behavior of a viscoelastic material as the authors also show, and it is certainly ok to provide this derivation. But the theory that is used in the end is fluid and not being clear here can put the reader on a wrong track.

The reviewer is correct: while microscopic description of the tissue may be more complex, on the time scales of interest, cellular flow is effectively viscous. Yet, we feel that it is important to note that “effectively viscous behavior” does not mean that the tissue is a simple viscous fluid, which it is not. (Note point #9 illustrates possible misunderstandings along this line of thought.) Our reference to the underlying visco-elastic model was intended to communicate that point and our derivation in Appendix 2: Mathematical model of tissue flow”relates the longitudinal and transverse effective viscosities to the shear and bulk elastic moduli and the stress relaxation rate. We have however made corrections suggested by the reviewer.

19) In Figure 3 a coordinate independent, covariant expression of the dynamic equations is presented. The equations given in the appendix are not covariant but depend on a coordinate system. It is therefore unclear whether the continuum theory applied to the problem is covariant or not. This should be clarified. Furthermore, the finite element method used is also not clear with respect to this point and not sufficiently explained. Do the results depend on the choice of the finite element discretization used or not? If not, has this been checked? If yes, what does this mean? We understand that the authors do not want to go into notational complexities here but the conceptual approach used should be clear.

The equation presented in Figure 3 shows a general expression of the continuum model in terms of Laplace operator, divergence and gradient, which is valid on flat but also curved surfaces. Of course, these operators need to be correctly implemented (on a curved surface), which in our case is accomplished computationally via finite element method on a triangulated surface. For the derivation of our equation in the Appendix 2, however we assumed for simplicity a flat configuration, which greatly simplifies notation, and helps the reader to develop a better intuition. In the Materials and methods section we now also explain how we tested accuracy of our finite element solver and dependence on the discretization used. In subsection “Finite element implementation”, we added:

“To benchmark the quality of our FEM solver, we implemented the equation of motion on a sphere and tested results for known solutions, which confirmed our solver works within the expected numerical accuracy. To test dependence on discretization used, we tested our results using different meshes at varying mesh sizes and found good agreement with all test cases.”